# Non-Destructive Possibilities of Thermal Performance Evaluation of the External Walls

**DOI:** 10.3390/ma14237438

**Published:** 2021-12-04

**Authors:** Henryk Nowak, Łukasz Nowak

**Affiliations:** Faculty of Civil Engineering, Wroclaw University of Science and Technology, Wybrzeże Wyspiańskiego 27, 50-370 Wrocław, Poland; henryk.nowak@pwr.edu.pl

**Keywords:** external walls, thermal measurements, R-value, thermal resistance, temperature-based method, heat flow meter method, infrared thermography method

## Abstract

Identification of the actual thermal properties of the partitions of building enclosures has a significant meaning in determining the actual energy consumption in buildings and in their thermal comfort parameters. In this context, the total thermal resistance of the exterior walls (and therefore their thermal transmittance) in the building is a major factor which influences its heat losses. There are many methods to determine the total thermal resistance of existing walls, including the quantitative thermography method (also used in this study). This paper presents a comparison of the calculated total thermal resistance values and the measured ones for three kinds of masonry walls without thermal insulation and the same walls insulated with expanded polystyrene boards. The measurements were carried out in quasi-stationary conditions in climate chambers. The following three test methods were used: the temperature-based method (TBM), the heat flow meter method (HFM) and the infrared thermography method (ITM). The measurement results have been found to be in good agreement with the theoretically calculated values: 61% of the measured values were within 10% difference from the mean value of total thermal resistance for a given external wall and 79% of the results were within 20% difference. All of the used measuring methods (TBM, HFM and ITM) have proven to be similarly approximate in obtained total thermal resistances, on average between 6% and 11% difference from the mean values. It has also been noted that, while performing measurements, close attention should be paid to certain aspects, because they can have a major influence on the quality of the result.

## 1. Introduction

Buildings play an important role in achieving sustainable development goals, among others aimed at reducing greenhouse gas emissions, by using renewable energy sources and reducing annual energy consumption. To achieve this goal, one should use known ways of solar energy (passive and active) utilization, low-temperature geothermal energy, highly efficient heating, ventilation and air conditioning (HVAC) equipment and domestic hot water production equipment. However, the key is to achieve the optimal energy performance of the building by significantly limiting heat losses through its thermal envelope, mainly through its exterior walls.

Apart from the behavior of the users, the specific energy consumption in a building strongly depends on the thermophysical properties of its partitions (building envelope elements) [1,2]. The thermal envelope of the building, especially its exterior walls, is an important factor in shaping its annual heat balance and thermal comfort parameters. The basic quantities that characterize the thermal properties of walls are thermal resistance (*R*), total thermal resistance (*R_tot_*) and the thermal transmittance (*U*). The accuracy with which these quantities are determined can significantly affect the accuracy of the actual and forecasted energy consumption of buildings [3,4]. In situ measurements of the U-value have been performed since the early 1980s [5,6,7]. The measurements were found to be very useful for determining the actual thermal performance of the exterior walls in existing buildings, especially since the values determined on-site in buildings in the actual conditions would often significantly differ from the calculated ones. In the subsequent years, several experimental and computational studies aimed at determining the total thermal resistance of walls, using the approach for heat flow steady state, including reviews of the measurement methods [8,9], methodology [10,11,12,13], the use of in situ methods [2,14,15,16,17] and unsteady state conditions [18,19,20,21], were carried out.

There are many different theoretical and practical methods of determining the total thermal resistance of the exterior walls. On the one hand, the simplest way to determine the total thermal resistance of building enclosures consists of performing calculations consistent with ISO 6946 [22], where formulas can be found for the calculations depending on the structural system of the building enclosure. On the other hand, in practice, there often occur discrepancies between the tabular and actual thermal properties of the individual layers of a building partition. This applies mainly to the thermal conductivity coefficient of the particular layers and their density (and therefore their heat capacity) [12,23,24]. As a result, it often becomes necessary to carry out additional measurements, e.g., of the thickness of the individual layers or the density of the materials (destructive methods). Calculations in accordance with ISO 6946 are mainly performed when designing exterior building enclosures for new buildings. Whereas when it is necessary to determine the total thermal resistance through in situ measurements in existing buildings, the international ISO 9869-1 standard [25], recommending the use of the heat flow meter method (HFM), is applied. Although this standard has several advantages, two major problems which can be encountered when using it, i.e., the long duration of the measurements due to unstable boundary conditions and the questionable accuracy of the measurements, were indicated in [26,27]. Standard ISO 9869-1 recommends a minimum measurement duration of 72 h (a multiple of the full 24 h) and that the total thermal resistance values from the two last measurement days should not differ by more than 5%. Moreover, using this standard, one can analyze measurement data by the average method and the dynamic method (the latter yields more accurate measurement results). A new addition to the documents related to in-situ measurement of thermal resistance of building partitions is the ISO 9869-2 standard, which introduces the infrared camera method [28]. In issues related to heat transfer through building partitions and the determination of their thermal resistance, a very important issue is to take into account the variability of material moisture and its transfer [29,30]. However, this research problem was not within the scope of this article.

Another method of determining the total thermal resistance of a building enclosure, and thus determining the latter’s U-value, is the generally known and well-established infrared thermography method (ITM). For many years, infrared thermographic surveys of buildings have been a precise and quick quality evaluation method [31,32,33], mainly used to locate defects in the thermal envelope of buildings [34,35,36] with presentations of basic and research issues and their applications [37,38]. Using this method, thermal bridges can be identified as caused by discontinuities or absence of thermal insulation [39,40,41,42], damp patches in building fabric elements [43,44,45] and places of air infiltration through the envelope [46,47,48]. Infrared thermography offers great possibilities to qualitatively assess the heat losses of different types of buildings and to analyze them to fully solve the problem of the energy performance of buildings [49]. This method is most often used as passive thermography, that is, as a method to investigate the building fabric as-is, without controlled outside interference and thermal stimulation of its thermodynamic state [35,38,50,51]. It can also be used as active thermography, i.e., with controlled outside temperature interference in the thermal state of the investigated building fabric [52,53,54,55] and inverse contrast in NDT materials’ research [56,57]. Furthermore, through thermographic measurements, one can also determine the physical quantities characterizing the investigated building enclosure. Taking this into account, thermal imaging studies have been used to quantify external walls, including the U-value of walls [58,59], heat flux density [60,61,62,63] and to compare various measurement and calculation methods [64,65]. As part of the present study, quantitative thermographic measurements were carried out in climatic test chambers using passive ITM to determine the total thermal resistance of the investigated building enclosure elements, and therefore their thermal transmittance.

For many years, building fabric elements also have been investigated in climate chambers, most often in quasi-stationary conditions. Such investigations deal with many problems related to building physics, energy performance of buildings, including all kinds of building fabric and thermal insulation [66,67,68,69], and the effect of ventilation on thermal comfort and indoor air quality (IAQ), particularly in office buildings [70]. The advantage of tests conducted in climate chambers over in situ tests in buildings is that they are independent of the weather conditions and therefore can be conducted practically all year round, whereby they are not limited to only the heating season and can be carried out for a set difference in air temperature between the two sides of a building enclosure. Practically, this means that tests conducted in climate chambers are more productive as regards the number of building fabric elements tested per year.

Taking into account the above, the main aim of the present research was to compare the effectiveness of the various methods of testing masonry walls by comparing the total thermal resistance values calculated and measured under quasi-stationary conditions in climate chambers for three types of masonry walls without thermal insulation and for the same walls insulated with expanded polystyrene. The same difference in air temperature between the walls’ two sides was maintained for each of the tested walls. Three testing methods were used and evaluated with respect to the accuracy of the measurement results they produced. The methods were as follows: the temperature-based method (TBM), the heat flow meter method (HFM) and the infrared thermography method (ITM), which is also referred to in the literature as the infrared thermography technique (ITT). The purpose of the investigations was also to evaluate the effectiveness of the methods and to highlight the key factors that influence the accuracy and correctness of the measurement.

## 2. Thermal Resistance Assessment for Building Enclosure Parts

### 2.1. Calculation Method

In the design stage, an assessment of the thermal properties of a building enclosure is usually based on calculations of the unidirectional heat transmission process described in ISO 6946 [22]. Heat transmission through a building enclosure consists of the absorption of heat from the environment with a higher temperature (usually a heated room) by the building enclosure’s surface, the flow of the heat through the building enclosure and the transfer of the heat from the building enclosure’s other surface to the air with a lower temperature, and is defined by the partition’s total thermal resistance—*R_tot_* ((m^2^K)/W), determined from the relation:(1)$$R_{tot}=R_{si}+R+R_{se}$$
where:

$R_{si}$, $R_{se}$—surface thermal resistances, on the internal and external side respectively, assumed or calculated according to Section 2.3.1 ((m^2^K)/W), 

R—the design thermal resistances of the partition layers ((m^2^K)/W), determined from the relation:(2)$$R=\overset{n}{\underset{i=1}{\sum }}\frac{d_{i}}{\lambda_{i}}$$
where:

*d_i_*—the thickness of a given material layer (m),

*λ_i_*—the thermal conductivity coefficient of the given material layer (W/(mK)).

The above method is commonly used in newly designed buildings to evaluate their energy performance and also to confirm that they meet the energy conservation requirements specified by the binding regulations. However, sometimes, there can arise the need to evaluate the thermal performance of building fabric through measurements, i.e.,

In the case of a new building, when there is a suspicion as to the thermal performance of the materials used when confronted with the design specifications.In the case of an existing building, to assess its technical condition or when there is no relevant information in its design documents.

### 2.2. In Situ Measuring Methods

The thermal properties of building fabric elements are usually measured when a quantitative or qualitative evaluation of their thermal performance is needed. Although thermographic surveys are mainly used for quality assessments of building enclosures, they can also be the basis for a quantitative evaluation of their thermal properties, even without simultaneously carrying out additional measurements of other quantities. Even though qualitative thermographic tests are sufficient for the purpose of assessing the condition of the thermal insulation in building enclosures, for many years in the literature on the subject, attempts have been made to quantitatively evaluate building enclosures through thermographic tests. The following three main methods for solving this problem can be distinguished:Method 1—the temperature-based method (TBM), consisting of measuring air temperatures and the building enclosure’s surface temperatures on both sides.Method 2—the heat flow meter method (HFM), consisting of measuring heat flux densities and the surface temperatures of the building enclosure on both sides.Method 3—the infrared thermography method (ITM), consisting of measuring air temperatures and the building enclosure’s surface temperatures on both sides (similarly as in Method 1) and performing radiometric measurements on both sides of the building enclosure: this method is precise, but labor-intensive.

#### 2.2.1. Method 1—The Temperature-Based Method (TBM)

The U-value of a building partition can be determined by measuring the appropriate temperatures and calculating its total thermal resistance, expressed by the relation:(3)$$R_{tot}=\frac{R_{si}\left(T_{i}-T_{e}\right)}{\left(T_{i}-T_{si}\right)}$$
or the relation:(4)$$R_{tot}=\frac{R_{se}\left(T_{i}-T_{e}\right)}{\left(T_{se}-T_{e}\right)}$$
where:

$R_{si}$, $R_{se}$—internal surface thermal resistance and external surface thermal resistance ((m^2^K)/W),

$T_{i}$, $T_{e}$—air temperature on respectively the internal and external side of the tested building partition (K),

$T_{si}$, $T_{se}$—the temperature of respectively the internal and external surface of the tested building partition (K).

Temperature sensors should be attached to a building enclosure in a way that ensures proper contact with the latter using, e.g., thermal paste or adhesive tape. Furthermore, they should be so located that air temperature measurements are not exposed to disturbing factors (far from heat sources, windows and air diffusers). Temperature measurements should be conducted for such a period of time that a time window of about 72 h with visible building enclosure surface and air temperature stabilization on both sides of the tested building enclosure could be selected from this period.

#### 2.2.2. Method 2—Heat Flow Meter Method (HFM)

By measuring heat flux density, one can obtain a more accurate result as large fluctuations in air temperature readings can be partially eliminated in this way. In this case, the thermal resistance can be determined from the formula:(5)$$R=\frac{\left(T_{si}-T_{se}\right)}{q}$$
and then, total thermal resistance can be calculated from the formula:(6)$$R_{tot}=R_{si}+R+R_{se}$$
where:

$T_{si}$, $T_{se}$—the temperature of respectively the internal and external surface of the tested building enclosure (K),

q—the measured heat flux flowing through the building enclosure (W/m^2^),

$R_{si}$, $R_{se}$—the calculated surface thermal resistances on respectively the internal and external side of the building partition ((m^2^K)/W).

#### 2.2.3. Method 3—Infrared Thermography Method (ITM)

In the case of this method, the determination of thermal resistance performed as in Method 1 should be supplemented with radiometric measurements of the mean radiation temperature in the external environment and the mean radiation temperature in the room. This is because the formulas describing the unidirectional steady heat flow through building enclosures were derived assuming that the external air temperature and the mean external environment radiation temperature (representing the influence of the thermal radiation from the nearest surroundings of buildings) are equal. This assumption is a considerable simplification as a complex heat exchange occurs on the boundary surfaces of a building enclosure so that the heat flux density on a given surface of the building enclosure is equal to the sum of the densities of the heat fluxes transferred through convection and radiation. This means that the resultant heat flux proceeds from the building enclosure’s external surface via radiation towards ambient radiation temperature, *T_r_*, and via convection towards air temperature, *T*. The temperature, being a total thermal rating index of a physical environment, taking into account the radiation temperature of the surroundings and the air temperature, is known as operative temperature, *T_op_*, in environmental physics and is expressed by the formula:(7)$$T_{op,i}=\frac{h_{ci}T_{i}+h_{ri}T_{ri}}{h_{ci}+h_{ri}}$$
or the relation:(8)$$T_{op,e}=\frac{h_{ce}T_{e}+h_{re}T_{re}}{h_{ce}+h_{re}}$$
where:

$T_{i}$, $T_{e}$—air temperature on respectively the internal and external side of the building partition (K),

$T_{ri}$, $T_{re}$—mean radiation temperature on respectively the internal and external side of the partition (K),

$h_{ci}$, $h_{ce}$—the convection heat transfer coefficient on respectively the internal and external side of the partition (W/(m^2^K)),

$h_{ri}$, $h_{re}$—the radiative heat transfer coefficient on respectively the internal and external side of the partition (W/(m^2^K)).

Hence, the total thermal resistance of a building partition can be calculated from the relation:(9)$$R_{tot}=\frac{\left(T_{op,i}-T_{op,e}\right)}{q}$$
where:

q—the heat flux flowing through the building enclosure (W/m^2^).

### 2.3. Surface Thermal Resistances

#### 2.3.1. Simplified Method

The assumption of the correct value of surface thermal resistances (*R_si_* and *R_se_*) on both sides of the tested building enclosure has a noticeable effect on the calculated values of its total thermal resistance (*R_tot_*) and thermal transmittance (*U*). The European standard ISO 6946 [22] in its Table 7 suggests that *R_si_* = 0.13 m^2^K/W and *R_se_* = 0.04 m^2^K/W for an external partition and for horizontal heat flow.

#### 2.3.2. Calculation—Surface Temperatures and Air Movement Velocities Are Known

In real situations, depending on weather conditions, surface resistances *R_si_* and *R_se_* can diverge from the above values. In such cases, a more accurate method can be the one described in Appendix A of the ISO 6946 standard [22], according to which one can use the actual wind speed to obtain the value of the convective heat transfer coefficient, *h_c_*. In the research, it was assumed that the air movement caused by the fans of heating/cooling units in the climate chambers was an equivalent of wind, and because of that, coefficients *h_ci_* and *h_ce_* for both sides of the tested walls will be calculated in that way. The air movement was measured on, respectively, the internal side and the external side at the tested building enclosure by means of anemometers. Then, surface thermal resistance *R_si_* and *R_se_* should be determined for both sides for the horizontal heat flux flow, using the formulas:(10)$$R_{si}=\frac{1}{h_{i}}=\frac{1}{h_{ci}+h_{ri}}$$
(11)$$R_{se}=\frac{1}{h_{e}}=\frac{1}{h_{ce}+h_{re}}$$
where:

$h_{ci}$*,*$h_{ce}$—the convective heat transfer coefficient on respectively the internal side and the external side, calculated from the formulas:(12)$$h_{ci}=4+4\nu_{i}$$(13)$$h_{ce}=4+4v_{e}$$
where:

$\nu_{i}$, $\nu_{e}$—wind speed adjacent to the surface of respectively the internal and the external side (m/s),

$h_{ri}$, $h_{re}$—the radiative heat transfer coefficient on respectively the internal side and the external side, calculated from the formulas:(14)$$h_{ri}=\varepsilon h_{ri,0}=\varepsilon 4\sigma T_{mi}^{3}$$(15)$$h_{re}=\varepsilon h_{re,0}=\varepsilon 4\sigma T_{me}^{3}$$
where:

ε—surface emissivity into a half-space, assumed as equal to 0.9 (–),

$h_{ri,0}$, $h_{re,0}$—a black body radiation heat transfer coefficient (W/m^2^K),

σ—the Stefan–Boltzmann constant equal to 5.67 × 10^−8^ (W/(m^2^K^4^)),

$T_{mi}$, $T_{me}$—mean thermodynamic temperatures of the surfaces (K).

#### 2.3.3. Calculation—Air Temperatures, Surface Temperatures and Heat Flux Density Are Known

In the case when it is possible to carry out additional surface temperature measurements on both sides of the tested building enclosure and to determine the heat flux, the surface thermal resistances can be experimentally determined in accordance with the formulas:(16)$$R_{si}=\frac{\left(T_{i}-T_{si}\right)}{q}$$
(17)$$R_{se}=\frac{\left(T_{se}-T_{e}\right)}{q}$$
where:

$T_{si}$, $T_{se}$—the temperature of respectively the internal and external surface of the analyzed element (K),

$T_{i}$, $T_{e}$—the air temperature on respectively the internal side and the external side (K),

q—the heat flux flowing through the building enclosure (W/m^2^).

The above methods were used in the analyses carried out by the authors, but this does not exhaust the subject. There are many methods of estimating radiative heat transfer coefficients, *h_r_*, and convective heat transfer coefficients, *h_c_*. A comprehensive analysis of this subject can be found in [63].

## 3. Materials and Methods

### 3.1. Tested Elements

The research was carried out for a part of the building enclosure, namely the external walls. Three different materials were tested: aerated concrete blocks, ceramic bricks and concrete blocks, as these are some of the most commonly used materials in residential buildings in Poland. The homogeneous tested walls were divided into two groups: uninsulated and insulated. Wall A was made of 240 mm × 240 mm × 590 mm class 600 H + H aerated concrete blocks laid in Baumit ThermoMörtel 50 insulating mortar. In the case of the insulated version of wall A, 10 cm thick EPS (expanded polystyrene) boards were glued to it on the cooler side and then covered with fiberglass-reinforced mineral render. Wall B was made of 60 mm × 120 mm × 250 mm solid ceramic bricks laid in cement-lime mortar. In the case of the insulated version of wall B, 10 cm thick EPS boards with the same properties as before were glued to it on the cooler side and then covered with fiberglass-reinforced mineral render. Wall C was made of 120 mm × 250 mm × 380 mm solid concrete blocks laid in cement-lime mortar. In the case of the insulated version of wall C, 10 cm thick EPS boards were also used. After building each wall, a time of 4–6 weeks was used to season the construction and to get rid of construction moisture. During that time, the doors of the climate chambers were left open. The basic material specifications of the walls are presented in Table 1. The given thermal conductivity coefficients are declared values and were taken from producer technical specifications, provided by the construction material warehouse or from tabular data from Polish technical standards [71,72,73]. These values were not experimentally verified (measured) by the authors.

### 3.2. Test Setup

Measurements of the thermal resistance of selected external walls were conducted in a sleeve between two connected climate chambers, as shown in Figure 1. The major specifications of the set are as follows:The warm chamber (on the left side, blue one): capacity 30 m^3^, internal dimensions 3.0 m × 4.0 m × 2.5 m, temperature range −30–+80 °C and relative humidity range 10–95%.The cold chamber (on the right side, grey one): capacity 30 m^3^, internal dimensions 3.0 m × 4.0 m × 2.5 m, temperature range −40–+85 °C and relative humidity range 10–95%.

The tested wall would be placed in a sleeve connecting the two climate chambers, in which conditions simulating the behavior of building enclosures in the heating season were maintained, that is:The warm chamber with activated controlled air temperature θ_i_ = +20 °C and relative humidity φ_i_ = 50%.The cold chamber with activated controlled air temperature θ_e_ = −10 °C and uncontrolled relative humidity φ_e_ in the area of 30%.

All homogenous building enclosures were tested from the instant the temperature settings in the chambers were activated (at the instant of activation the air temperature in the chambers was equal to the temperature in the laboratory room, approximately 20–25 °C) until the temperatures in the chambers and the heat fluxes stabilized. Then, the measurements were conducted for at least 7 days. All the enclosures were positioned along the axis of the sleeve connecting the two climate chambers. The erected wall was left for the mortar to set. A schematic diagram of the sensor connection on both sides of the tested building enclosures is shown in Figure 2. Then, sensors were placed on both sides, and as part of the measurements, the following were registered:Air temperatures in the chambers by means of Ahlborn FHA646-E1 and FHA646-E1C T/RH (temperature and relative humidity) sensors with an accuracy of ±0.2 °C, in the range of 5–60 °C.Air temperature on the warm chamber side with the FLIR P65 thermal imaging camera with an accuracy of ±2 °C or ±2% of the measured value.Air movement velocities in the chambers with the help of the Ahlborn FVA935-TH4K2 TA5O anemometer with an accuracy of ±0.04 m/s + 1% of the measured value, in the range of 0.08–2 m/s, and the FVA605-TA5O anemometer with an accuracy of ±1.5% of the measured value, in the range of 0.15–5 m/s.Wall surface temperatures by means of class 2 NiCrNi thermocouples with an accuracy of ±2.5 °C or ±0.0075 multiplied by the measured value, in the range of −45–+200 °C, and a FLIR P65 thermal imaging camera on the warm chamber side.The density of the heat flux penetrating through the building enclosure with Ahlborn FQA 150–2 m with an accuracy of 5% at 25 °C.

All the Ahlborn sensors were connected to Ahlborn Almemo data loggers (2690-8, 2890-9 or 5690-2M09). In addition to this, a Hukseflux TRSYS01 system, dedicated to measuring the thermal resistance of building enclosures, with an accuracy of ±3% and the operating range of −30–+70 °C, which doubled the surface temperature measurements on both sides of the building enclosure and the density of the heat flux penetrating through the latter, was connected. A photo of one of the measurements carried out is shown in Figure 3.

### 3.3. Analyzed Methods of Determining Thermal Resistance for Selected Walls

As part of the investigations, the following methods:A computational method (Method 0a),A computational-measurement method (Method 0b),Measurement methods (Methods 1, 2a, 2b, 3a and 3b),were compared in determining the thermal resistance of 6 different building enclosures (walls A, B and C in their insulated and uninsulated versions). The methods differed from each other in (among other things):The source of the building partition parameters used in calculations—Method 0a is based solely on theoretical calculations using tabular material and surface resistance (*R_si_* and *R_se_*) values, Method 0b supplements such data with calculated surface resistances and the other methods (1–3b) use measured surface resistances.The parameters which need to be measured in order to obtain the total thermal resistance value—in Methods 1, 3a and 3b, the result is determined on the basis of only air and wall surface temperature measurements, whereas in Methods 2a and 2b, it is determined using the measured wall surface temperatures and heat flux densities.The way the required parameters are measured—in Methods 2a and 2b, thermocouples are used to measure wall surface temperatures and T/RH sensors are used to measure air temperatures, whereas in Methods 3a and 3b, both air and wall surface temperatures on the warm chamber side are measured by a thermal imaging camera.The kind of measuring equipment—in Method 2a, universal temperature sensors and heat flux density measuring sensors are used, whereas in Method 2b, a system dedicated to measuring the thermal resistance of building enclosures is used.The way of averaging the measured wall surface temperatures—in Method 3a, the surface temperatures measured by a thermal imaging camera are averaged on the basis of three random thermograms obtained from two spot measurements taken on the wall (six measurements in total), whereas in Method 3b, surface temperatures are averaged on the basis of three random thermograms based on the average for a 25 cm × 40 cm area.

All the above methods are compared in Table 2, which also includes information on how the components of total thermal resistance (*R_tot_*), i.e., surface thermal resistances (*R_si_* and *R_se_*) and thermal resistance (*R*) of the wall layers, were determined. Since the values of the thermal conductivity coefficients for the materials used (aerated concrete, solid ceramic bricks, concrete blocks, masonry mortar, expanded polystyrene and render) were taken from the specification tables and not verified by the authors, Methods 0a and 0b were not treated as references for the other methods, but on equal terms with the latter.

### 3.4. Air and Surface Temperatures from Thermal Imaging Camera (Methods 3a and 3b)

In this case, the temperatures were calculated as the average of readings from three randomly chosen thermograms recorded in the already stabilized heat flow period, selected from the same 72 h window as the measurements taken by the other sensors (i.e., heat flux density and surface temperature). In both methods, the air temperature was assumed as equal to the temperature of an object with a slight heat capacity and a high surface emissivity, placed at a certain distance from the tested wall. Therefore, the temperature of a spot on a black sheet of paper, denoted as SP01 on the thermogram, was read.

In Method 3a, the wall surface temperature as an average of six values was calculated. That means two spot readings from three random thermograms (two measurements on three thermograms). The spots were randomly selected avoiding wall corners, joints, sensor cables and other places with disturbed temperature fields. The surface temperature reading spots are marked as SP02 and SP03 on the thermograms.

In Method 3b, the wall surface temperature value used in thermal resistance calculations was obtained as the mean surface temperature of a certain area of the wall (1 measurement on 3 thermograms). The areas were randomly selected, but as before, places with temperature field disturbances were avoided. This area covered several masonry elements and the joints between them. On average, it was a 25 cm × 40 cm area. The surface temperature areas are marked as AR01 on the thermograms.

## 4. Results

### 4.1. Surface Thermal Resistances on Both Sides of Tested Walls

In the case of the computational method, the surface thermal resistances *R_si_* and *R_se_* for the tested building enclosure should be assumed in accordance with Table 1. While the value of 0.13 m^2^K/W assumed as surface resistance, *R_si_*, on the warm chamber side did not raise any doubts, the authors had doubts regarding the value of *R_se_* to be assumed on the cold chamber side. By assumption, this chamber mimics the external environment, but it was found that the air movement velocity measured there was several times lower (see the next paragraph) than the one assumed in ISO 6946 ($\nu_{e}$ = 4 m/s) for the given *R_se_* values. Therefore, for further analyses on the cold chamber side, *R_si_* = 0.13 m^2^K/W was assumed as for a horizontal heat flow, but for a space within the building envelope (e.g., for an unheated space).

During one of the measurement sessions in the climate chambers, in which air temperatures θ_i_ = +20 °C and θ_e_ = −10 °C were maintained in respectively the warm chamber and the cold chamber, the air movement velocities measured by the anemometers on average amounted to:
In the warm chamber $\nu_{i}$ = 0.56 m/s,In the cold chamber $\nu_{e}$ = 0.13 m/s.

Assuming that no surface temperature could be measured in this case (no surface temperature sensor is available), the mean thermodynamic temperatures of the two surfaces in Formulas (14) and (15) were set as the air temperatures in climate chambers, so T_mi_ = 293.15 K and T_me_ = 263.15 K. After that, the above air velocities were introduced into Formulas (12) and (13) and the surface thermal resistances were calculated from Formulas (10) and (11). Then, during measurements of the heat flux density and the air and surface temperatures for each tested wall, the actual surface resistances on both sides of the enclosure were determined from Formulas (16) and (17). All of the *R_si_* and *R_se_* values are presented in Table 3. The clear differences between the calculated surface resistances and the measured ones are due to the measurement conditions that differed from the real building enclosure service conditions. Additionally, the specific way in which air with set temperature is blown into climate chambers (different chamber designs), how closely the temperature sensors adhere to the analyzed wall surface and the heat flux density contribute to the differences.

### 4.2. Air Temperatures in Chambers and Wall Surface Temperatures

#### 4.2.1. Measurements with Thermocouples

Diagrams of the internal (warm) surface temperature and external (cold) surface temperature of the walls measured with thermocouples and air temperatures in the warm chamber and in the cold chamber during the measurements are shown in Figure 4 for the walls without insulation and in Figure 5 for the insulated walls.

From the start of the test (switching on the chambers), the temperature values began to approach the target temperature settings, i.e., θ_i_ = +20 °C and θ_e_ = −10 °C. The temperature values given below are the average of the temperatures in the 72 h time window (marked with a double-headed arrow in the diagrams) selected from the period of stabilized temperatures. In the case of wall A (Figure 4a), one can notice that the time in which the planned temperature settings were reached amounted to about 24 h, whereby the surface temperatures also stabilized. The latter on average was θ_si_ = +17.8 °C and θ_se_ = −6.8 °C. In the case of wall B (Figure 4b), the time in which the planned temperature settings were reached was similar (about 36 h), but the surface temperatures stabilized after about 96 h from the start, ranging on average to θ_si_ = +10.7 °C and θ_se_ = −0.6 °C. In the case of wall C (Figure 4c), the time in which the planned settings were reached amounted to about 36 h and the surface temperatures stabilized after 90 h, on average amounting to θ_si_ = +9.7 °C and θ_se_ = +0.4 °C. Then, the walls above were insulated with expanded polystyrene and tested again in the climate chambers. The temperature diagrams for insulated wall A are shown in Figure 5a. In this case, the time in which the planned temperature settings were reached was much shorter, slightly more than 6 h. The surface temperatures stabilized after less than 24 h, but probably due to the fact that the door to the warm chamber had been accidentally left open there are visible fluctuations in air temperature in the third 24 h of measurements. Therefore, it was decided to average the temperatures measured from the 72nd hour onwards, whereby θ_si_ = +19.5 °C and θ_se_ = −7.8 °C were obtained. The temperature diagrams for insulated wall B are shown in Figure 5b. In this case, the time to reach the planned temperature settings was disturbed by the abnormal operation of the cold chamber, in which the temperature stabilized as late as after 48 h. The surface temperatures averaged from the 84th hour onwards amounted to θ_si_ = +18.3 °C and θ_se_ = −8.1 °C. In the case of insulated wall C (Figure 5c), the time in which the planned temperature settings were reached was slightly longer, amounting to about 12 h. The surface temperatures stabilized relatively quickly (after less than 24 h), but for averaging purposes, the measurements from the 84th hour onwards were taken, whereby θ_si_ = +18.8 °C and θ_se_ = −7.5 °C were obtained.

#### 4.2.2. Thermograms of Tested Walls

While the walls were tested in the climate chambers, they were also observed by means of an infrared camera, which recorded data at certain intervals (usually at every 10 min) during the measurement period. The selected thermograms in Figure 6 show temperature field distributions in the period of stabilized temperatures (72 h period shown in Figure 4 and Figure 5) for the analyzed walls in their uninsulated version (Figure 6a,c,e) and insulated version (Figure 6b,d,f). Table 4 contains measured values for points SP01, SP02 and SP03, and area AR01 for all three randomly chosen thermograms. 

On the thermograms basis it can be, in addition, concluded that:Distinct drops in temperature were visible in the joints between the aerated concrete blocks in uninsulated wall A (Figure 6a).The joints between the blocks and bricks were practically invisible for insulated wall A (Figure 6b) and for uninsulated and insulated wall B (Figure 6c,d).No joints were visible due to the similar thermal conductivity of the mortar and the concrete blocks for uninsulated and insulated wall C (Figure 6e,f).

### 4.3. Heat Flux Density

Diagrams of the heat flux density measured by a 0.5 m × 0.5 m Ahlborn plate sensor for all the tested walls are shown in Figure 7 and Figure 8. The density of the heat flux flowing through uninsulated wall A (aerated concrete) is shown in Figure 7a. The heat flux increased to stabilize after about 24 h, ranging instantaneously from 17.0 to 24.8 W/m^2^. The average heat flux value in the thermal resistance calculation period amounted to about 21.0 W/m^2^. The diagram in Figure 7b is for uninsulated wall B. In this case, the heat flux increased noticeably longer and stabilized only after about 96 h at the mean value of 55.1 W/m^2^, ranging instantaneously from 50.9 to 59.2 W/m^2^. Figure 7c shows the heat flux for wall C. The heat flux increased to stabilize after about 48 h. However, because of the two visible interruptions in data logging (approximately the 80th and 160th measurement hour), the period from the 90th hour to the 156th hour was selected to determine thermal resistance. The heat flux density on average amounted to 57.7 W/m^2^, instantaneously ranging from 51.9 to 66.2 W/m^2^.

After the tested walls had been insulated, heat flux density noticeable decreased, as shown in Figure 8. For insulated wall A (Figure 8a), heat flux density stabilized after 72 h, ranging instantaneously from 4.5 to 9.9 W/m^2^ and on average amounting to about 7.1 W/m^2^. Additionally, an interruption in data logging between the 48th hour and the 66th hour of measurement is visible in the graph. The results for insulated wall B are presented in Figure 8b. In this case, the measurements were undisturbed, and the heat flux stabilized after about 84 h, ranging instantaneously from 6.6 to 9.6 W/m^2^, on average amounting to 7.7 W/m^2^. In the measurement of heat flux density (Figure 8c) for wall C, one can see distinct vertical heat flux fluctuations, probably due to the “rippling” of the plate sensor caused by heating, resulting in alternately better and worse adhesion of the sensor to the tested wall. The sensor was fixed to the wall with strong duct tape, but it is possible that the tape became unstuck, or due to the deformation of the sensor’s material (PTFE), it did not adhere the sensor properly to the surface of the wall. In the case of this building enclosure, the heat flux was averaged from the 84th hour to the 156th hour. In this period, heat flux density ranged instantaneously from 6.7 to 12.4 W/m^2^, on average amounting to 9.1 W/m^2^.

### 4.4. Thermal Resistance of Building Enclosures—Comparison of Methods

The thermal resistance of the tested walls was calculated on the basis of the values, measured within the 72 h interval marked in the diagrams on Figure 7 and Figure 8, and as a result, the total thermal resistance values are presented as bar charts in Figure 9 and in Table 5. The quantitative and percentage differences between the total thermal resistance results are clearly visible in the table.

It can be seen that, for some of the walls analyzed and the methods used, the results were different by more than 20% from the average value of total thermal resistance—such as wall C, Method 1, 2b, 3a and 3b, or wall B insulated, Method 2a and 3b. The probable reason behind that is presented in the Discussion Section. As it appears from Figure 9, quantitative differences in the results are more noticeable in the thermal resistances calculated for the insulated versions of tested walls, which may suggest that a thermal error or an inaccuracy in the execution of temperature and heat flux density measurements has greater consequences in this case.

The mean difference between the total thermal resistance results obtained for a given method and the average from all methods is shown in the last column in Table 6. When calculating the mean difference, one result most divergent from the other results yielded by a given method was neglected, as it was assumed to be due to an incorrectly performed measurement. The methods ordered from the one yielding results most consistent with the mean value obtained from all the methods are as follows: Method 0b—the *R_tot_* results differed on average by 3.6% from the mean value from all the methods, Methods 0a and 2b—by 6.1%, Method 3a—by 6.5%, Method 3b—by 7.7%, Method 2a—by 8.2% and Method 1—by 10.6%.

Then, in Table 6, for each of the methods of determining the total thermal resistance of the tested walls, the three results closest to the mean value from the measurements were marked with a green background and those farthest from the mean value were marked with a red background. If one regards closeness of the result to the mean value as indicative of the effectiveness of the method of determining the total thermal resistance, the methods should be ordered as follows: Method 0b—5 × the closest results, no farthest results,Method 0a—4 × the closest results, 1 × the farthest results,Method 3b—4 × the closest results, 2 × the farthest results,Method 2b—2 × the closest results, 3 × the farthest results.Method 2a—2 × the closest results, 4 × the farthest results.Method 3a—1 × the closest results, 4 × the farthest results.Method 1—no closest results, 4 × the farthest results.

## 5. Discussion

The measurement results and the total thermal resistances calculated from them showed, in the majority of cases, good agreement with the theoretically determined values. For the thirty measurements carried out using the five different methods, i.e., 1, 2a, 2b, 3a and 3b, 19 results differed by up to 10% from the average for a given building enclosure, 5 results differed by up to 20%, 5 results differed by up to 30% and only 1 differed by nearly 40% from the mean value. Several practical conclusions concerning the methods used to emerge from the extensive tests are given below:Method 0a—theoretical calculations and resistances according to Table 7, ISO 6946, is quite sufficient, in the authors’ opinion, to evaluate the thermal performance of the building enclosures being designed, since in this case, it is not possible to assess the actual surface thermal resistances (*R_si_* and *R_se_*) at the building enclosure surface.Method 0b—theoretical calculations supplemented with real measurements of air movement velocity at the building enclosure, used to calculate surface thermal resistances, yields total thermal resistances closest to the mean values.Method 1—Temperature-based method (TBM), in theory easy and relatively cheap to implement (only temperature sensors and data acquisition devices are needed), in practice turned out to be least accurate; nevertheless, it can still be regarded as acceptable—the mean error amounting to 10.6% and the maximum measurement deviation from mean value *R_T_* reaching no more than 23%. The accuracy of this method is most affected by the location of the sensor measuring temperature only in a particular spot, and so there is no certainty that this place of measurement is representative enough for the whole building partition.Method 2a—Heat flow meter method (HFM), based on measurements by a large (0.50 m × 0.50 m) heat flux density sensor, requires very good contact with the substrate, which is not always easy to achieve. In the above tests, contact of the sensor with the wall surface was ensured by circumferentially sticking it to the latter with strong tape. However, during several measurements, it was noticed that the sensor would undergo slight deformations under the influence of changes in the air and wall surface temperatures (it would bulge out and deflect from the surface). The conclusion is that a large amount of thermal paste should be used, but this would result in irreversible staining of the building enclosure surface.Method 2b—heat flow meter method (HFM), i.e., the use of a dedicated system of two small sets of heat flux density sensors and thermocouples in order to double the measurement of the parameters needed to determine thermal resistance, as the results show, is a slightly better solution—it is much easier to ensure proper constant contact of small (8 cm in diameter) heat flux density sensors with the substrate, using much less thermal paste, but it is still an invasive method.Methods 3a and 3b—Infrared thermography method (ITM) yielded qualitatively similar results as the ones yielded by the HFM methods, though Method 3a, based on averages from spot temperature measurements, was slightly more accurate. The great common advantage of the methods is their complete non-invasiveness towards the surface of the tested building enclosure.

When measuring the thermal resistance of building enclosures, the selection of surface thermal resistances, *R_s_*, strongly affects the quality of the results obtained, and in the authors’ opinion, resistances, *R_s_*, should also be determined using measurement methods. For this purpose, the air movement velocity at both building enclosure surfaces must be measured and the thermodynamic temperatures of the latter must be known.

Due to the wide range of tests, the following factors have been revealed that can randomly significantly affect the accuracy of the obtained results:Adhesion of sensors to the building enclosure surface: Even though small heat flux density sensors provide averages from a smaller area, it is definitely easier to attach them stably than in the case of large sensors. They should be attached using, e.g., thermal paste, but in a possibly small amount so that the thermal paste layer does not affect the measurement.Thermocouples are precise temperature sensors, but it is important that the cable connecting them with a data logger has damage-resistant insulation. The authors found silicone insulation to be the best as the fiberglass insulation did not perform well in the difficult measurement conditions (the cables would abrade, whereby the measurements would yield incorrect values).Air temperature: one can use a thin black sheet of paper to easily measure this temperature by means of an infrared camera, while taking a thermogram of the building enclosure surface.Thermographic measurement: if the building enclosures are relatively homogeneous (as in the analyzed cases), both the spot and area measurements of building enclosure surface temperature can be sufficiently accurate for the purpose of determining the thermal resistance of the building enclosure, but if the latter’s material is highly heterogeneous, the area temperature measurement is recommended.

## 6. Conclusions

The results of determining total thermal resistance in seven different ways based on four different methods (the theoretical method, TBM, HFM and ITM) for six different building enclosures with different thermal insulation properties, with or without a thermal insulation layer, made of different building materials, were presented above. The tests were carried out in climate chambers.

Considering that the values of the thermal conductivity coefficients of the materials used (aerated concrete, solid ceramic bricks, concrete blocks, masonry mortar, expanded polystyrene and render) were not verified by measurements by the authors, the theoretical methods (0a and 0b) were not treated as the references for the other methods, but on equal terms with them. For this reason, the results yielded by a particular method were compared with the average thermal resistances from all the methods.

The measurement results obtained for the purpose of calculating the thermal resistance of the building enclosures showed good agreement with the average thermal resistance, and in most cases, the differences did not exceed 10%, while 97% of determined resistances, *R_T_* (29 per 30 measurements), were within 30% of the difference.

The accuracy of the total thermal resistance measurement results obtained by the infrared thermography methods (for which the mean difference amounted respectively to 6.5% and 7.7%) is comparable with that of the results yielded by the heat flow meter methods (the mean difference amounted to 6.1% and 8.2%). The unquestionable advantage of the ITMs is their complete non-invasiveness, which means no problems with proper adhesion of sensors to the surface of building enclosures. Still, both the infrared thermography methods and the heat flow meter methods require the precise preparation of the test setup as errors affecting the accuracy of the methods can be easily committed. On average, the temperature-based method was characterized by worse accuracy than the other methods (the mean difference amounted to 10.6%), but unlike the other methods, the thermal resistance values yielded by it did not exceed 23% for any of the walls. When determining the total thermal resistance of a building enclosure, it is worth taking into account all the actual thermal transfer resistance values which may noticeably diverge from the ones assumed in ISO 6946 [22] for the internal and external surfaces of a building enclosure.

In the authors’ opinion, this research should continue, but the tests should be supplemented with measurements of the thermal conductivity coefficient of the tested materials by means of a dedicated device of the heat flow meter type (e.g., Fox 314 or Fox 600, etc.), which would provide a proper reference for the measurement methods. Moreover, the authors are inclined to attempt to determine the total thermal resistance by the dynamic method in accordance with ISO 9869-1 [27] and to consider the way in which heat transfer coefficients *h_r_* and *h_c_* are taken into account in the thermal resistance calculations according to the approaches available in the literature, extensively presented in [63].

## Figures and Tables

**Figure 1 materials-14-07438-f001:**
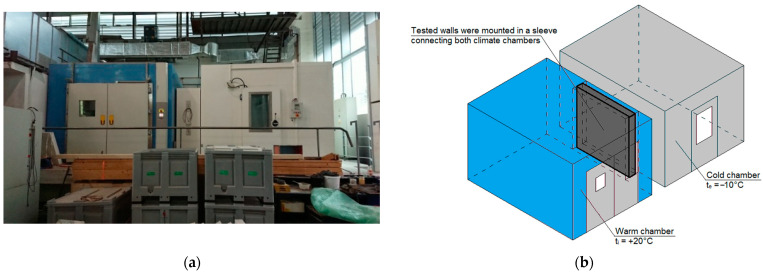
Test setup: (**a**) view of two climate chambers connected together, (**b**) locations of tested walls in sleeve connecting two climate chambers.

**Figure 2 materials-14-07438-f002:**
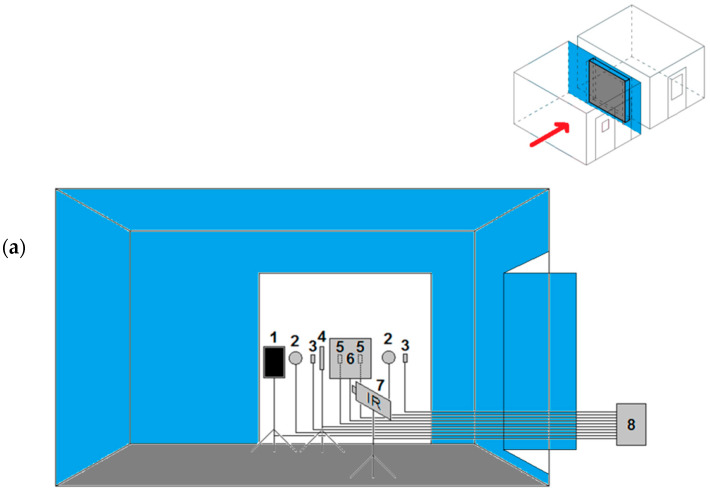
Schematic showing arrangement of sensors and measuring instrumentation on analyzed wall’s warm side (**a**) and on its cold side (**b**). Description of visible elements: 1—black sheet of paper (ambient temperature measurement by IR camera), 2—Hukseflux TRSYS01 system heat flux density sensors, 3—Hukseflux TRSYS01 system sensors measuring building enclosure internal surface temperature, 4—Ahlborn thermal anemometer (measuring internal air temperature and humidity), 5—Ahlborn sensors (thermocouples) measuring building enclosure surface temperature, 6—Ahlborn sensor (0.5 m × 0.5 m) measuring heat flux density, 7—FLIR P65 thermal imaging camera, 8—data loggers, 9—Hukseflux TRSYS01 system sensors for measuring building enclosure external surface temperature, 10—Ahlborn thermal anemometer (measuring external air temperature and humidity), 11—Ahlborn sensors (thermocouples) measuring building enclosure surface temperature.

**Figure 3 materials-14-07438-f003:**
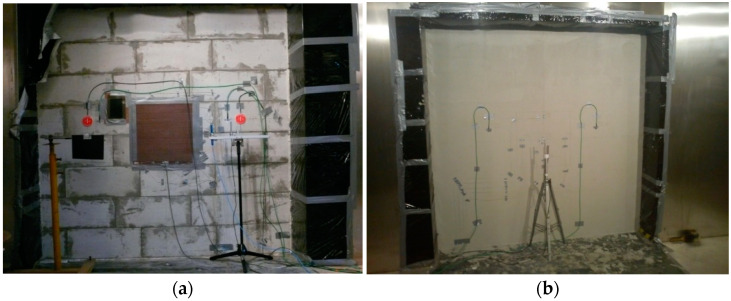
Photo of tested building enclosure (in this case, insulated wall A) with visible arrangement of sensors: (**a**) on warm chamber side, (**b**) on cold chamber side.

**Figure 4 materials-14-07438-f004:**
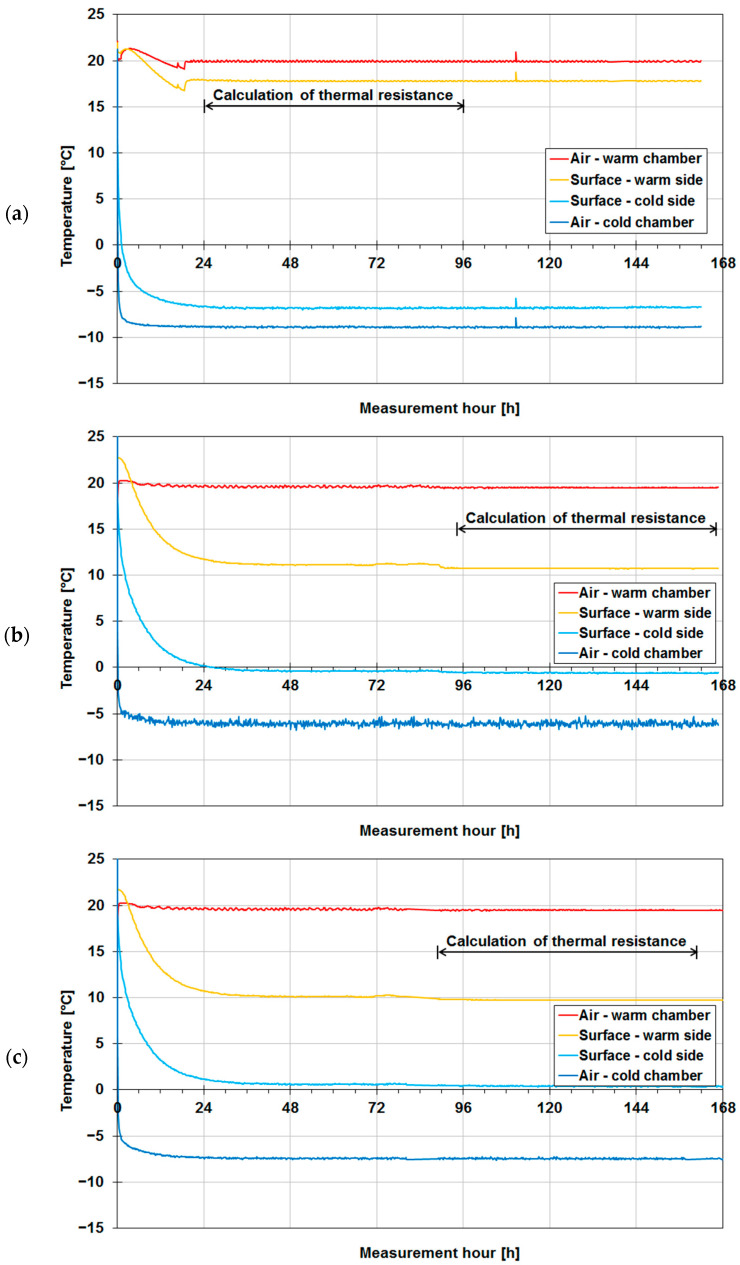
Air temperatures and temperatures of two surfaces of tested walls in their uninsulated version: (**a**) wall A made of aerated concrete, (**b**) wall B made of solid ceramic bricks, (**c**) wall C made of concrete blocks.

**Figure 5 materials-14-07438-f005:**
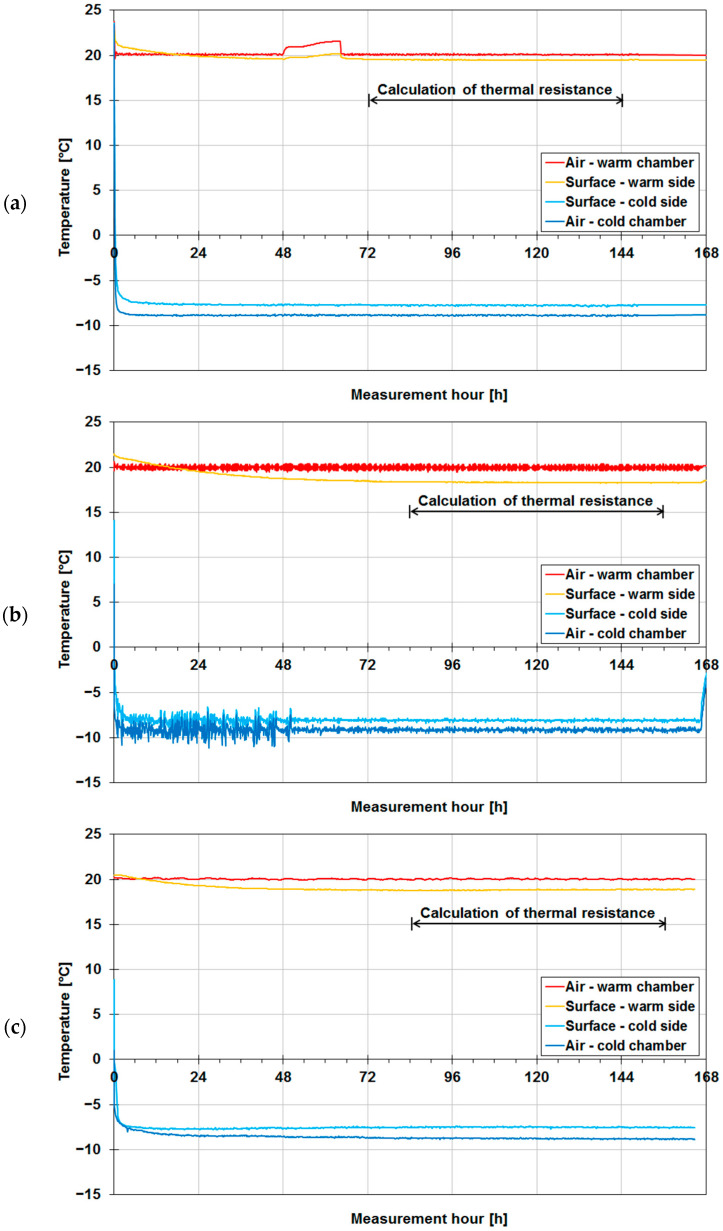
Air temperatures and temperatures of two surfaces of tested building partitions in their insulated version: (**a**) wall A made of aerated concrete, (**b**) wall B made of solid ceramic bricks, (**c**) wall C made of concrete blocks.

**Figure 6 materials-14-07438-f006:**
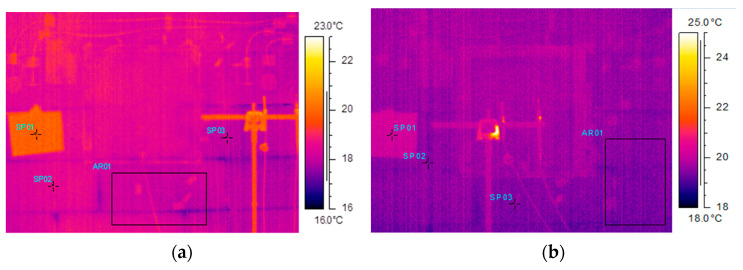
Comparison of thermograms for wall A in uninsulated version (**a**) and insulated version (**b**), wall B in uninsulated version (**c**) and insulated version (**d**) and wall C in uninsulated version (**e**) and insulated version (**f**).

**Figure 7 materials-14-07438-f007:**
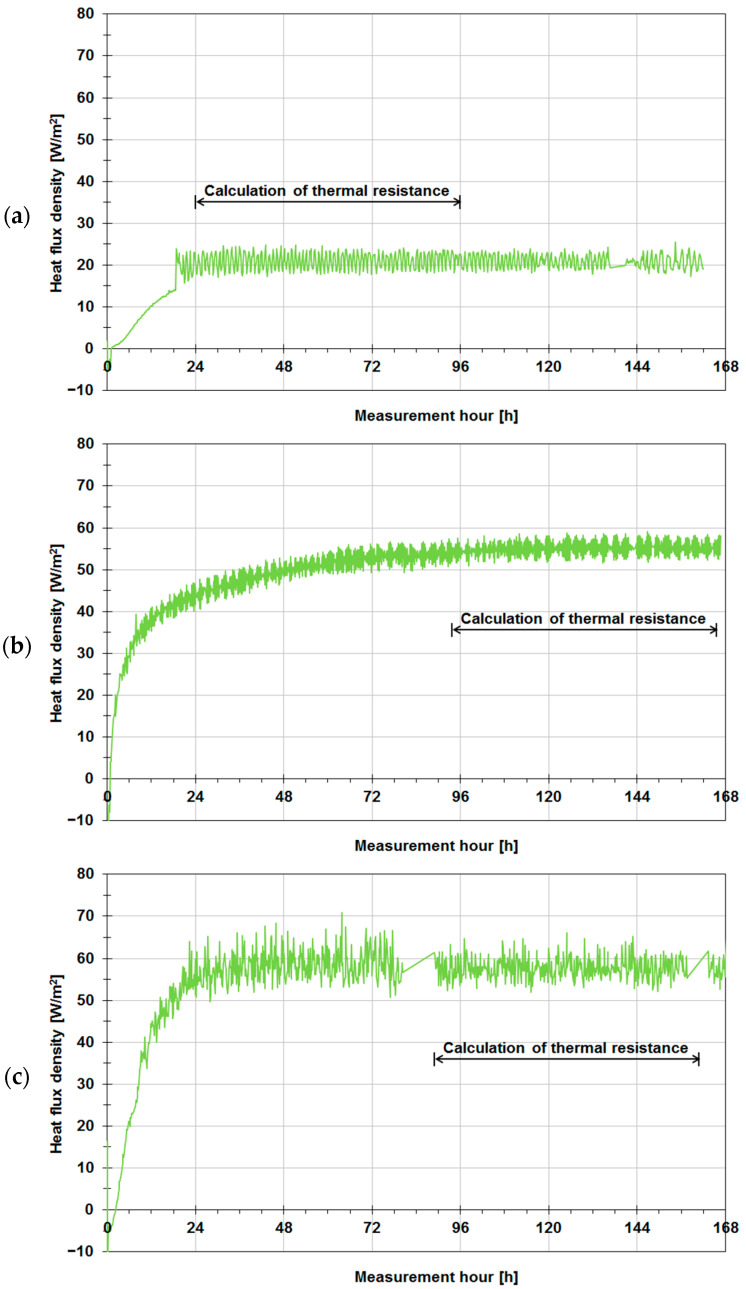
Density of heat flux flowing through tested building enclosures in their uninsulated version: (**a**) wall A made of aerated concrete, (**b**) wall B made of solid ceramic bricks, (**c**) wall C made of concrete blocks.

**Figure 8 materials-14-07438-f008:**
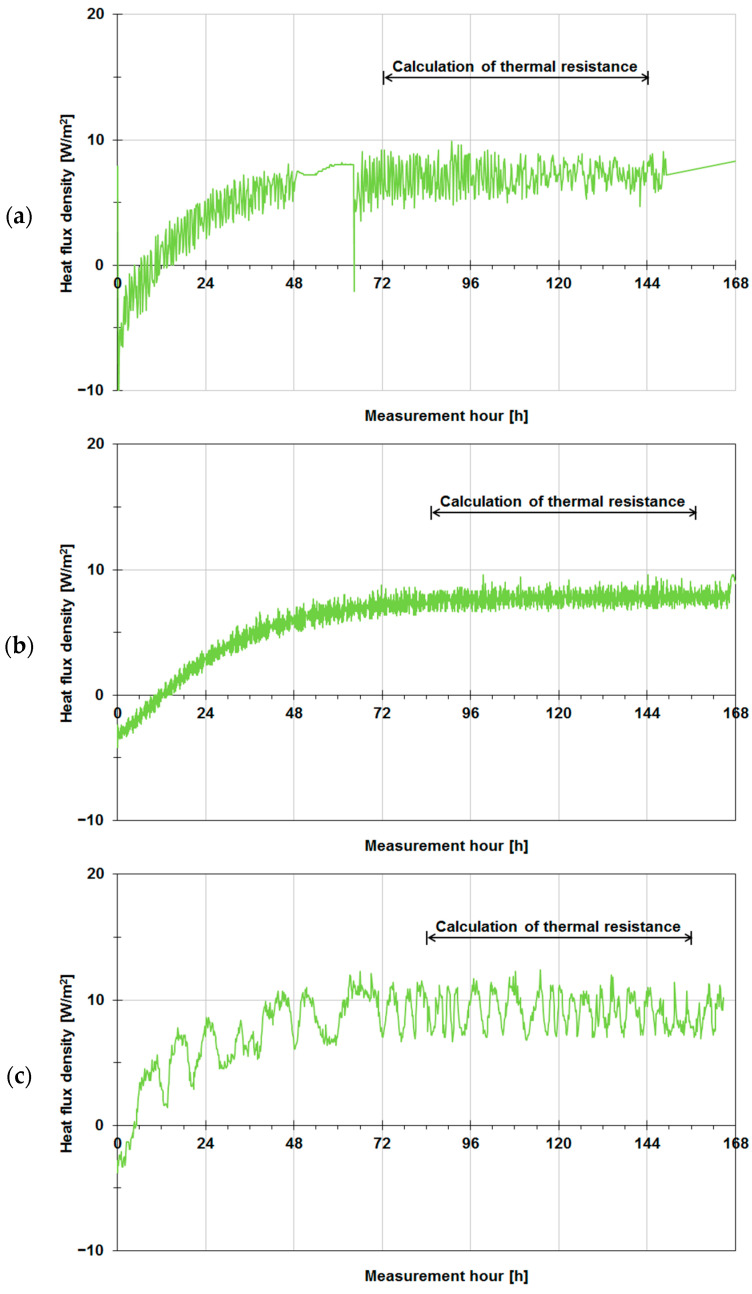
Density of heat flux flowing through tested building enclosures in their insulated version: (**a**) wall A made of aerated concrete, (**b**) wall B made of solid ceramic bricks, (**c**) wall C made of concrete blocks.

**Figure 9 materials-14-07438-f009:**
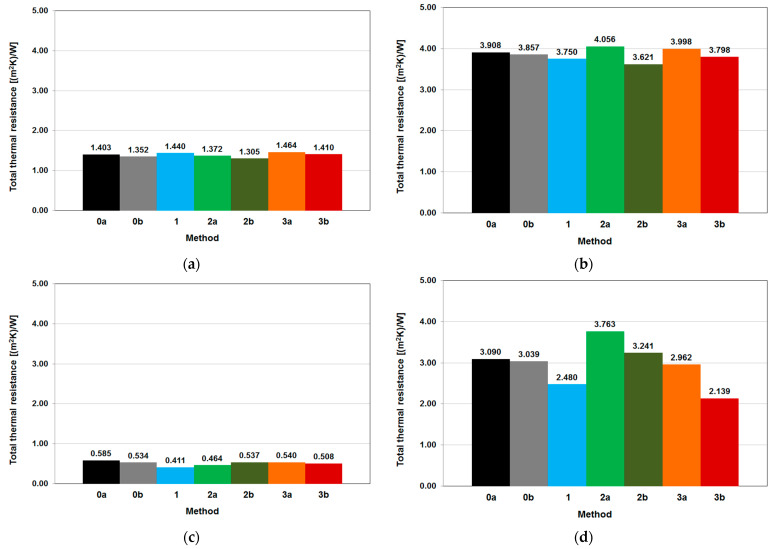
Comparison of thermal resistances measured using different methods for wall A in uninsulated version (**a**) and insulated version (**b**), wall B in uninsulated version (**c**) and insulated version (**d**) and wall C in uninsulated version (**e**) and insulated version (**f**).

**Table 1 materials-14-07438-t001:** Basic specifications of partitions tested in climate chambers.

Test Object	Material, Thickness (Innermost Layer First)	Declared Thermal Conductivity	Calculated R-Value
		W/mK	m^2^K/W
Wall A	Aerated concrete blocks, 24 cm	0.210 *	1.313
Wall A (insulated)	Aerated concrete blocks, 24 cm	0.210 *	3.818
	EPS boards, 10 cm	0.040 **	
	Fiberglass mesh-reinforced mineral render, 0.5 cm	1.000 ***	
Wall B	Solid ceramic brick, 25 cm	0.770 ***	0.495
Wall B (insulated)	Solid ceramic brick, 25 cm	0.770 ***	2.999
	EPS boards, 10 cm	0.040 **	
	Fiberglass mesh-reinforced mineral render, 0.5 cm	1.000 ***	
Wall C	Concrete blocks, 25 cm	1.000 ***	0.420
Wall C (insulated)	Concrete blocks, 25 cm	1.000 ***	2.925
	EPS boards, 10 cm	0.040 **	
	Fiberglass mesh-reinforced mineral render, 0.5 cm	1.000 ***	

* Taken from the technical data sheet for the material, ** provided by construction material warehouse, *** taken from tabular data from Polish technical standards.

**Table 2 materials-14-07438-t002:** Analyzed methods to determine the thermal resistance of building partitions.

Method of Determining Thermal Resistance of Building Enclosure			*R_si_*	*R*	*R_se_*	*R_tot_*
Name	Type	Description	m^2^K/W	m^2^K/W	m^2^K/W	m^2^K/W
0a	computational	Calculation using material data according to ISO 6946	ISO 6946, Table 7 (Section 2.3.1)	$R=\overset{n}{\underset{i=1}{\sum }}\frac{d_{i}}{\lambda_{i}}$ (Section 2.1)	ISO 6946, Table 7 (Section 2.3.1)	$R_{tot}=R_{si}+R+R_{se}$
0b	computational + measurement	Calculation using material data according to ISO 6946 + measurement of air movement velocity	ISO 6946, Appendix A (Section 2.3.2)	$R=\overset{n}{\underset{i=1}{\sum }}\frac{d_{i}}{\lambda_{i}}$ (Section 2.1)	ISO 6946, Appendix A (Section 2.3.2)	$R_{tot}=R_{si}+R+R_{se}$
1	measurement	Measurement of air and wall surface temperatures and air movement velocity by thermocouples	ISO 6946, Appendix A (Section 2.3.2)	-	ISO 6946, Appendix A Section 2.3.2)	$R_{tot}=\frac{R_{tot,i}+R_{tot,e}}{2}$ where: $R_{tot,i}=\frac{R_{si}\left(T_{i}-T_{e}\right)}{\left(T_{i}-T_{si}\right)}$ (Section 2.2.1) $R_{tot,e}=\frac{R_{se}\left(T_{i}-T_{e}\right)}{\left(T_{se}-T_{e}\right)}$ (Section 2.2.1)
2a	measurement	Measurement of air and wall surface temperatures by thermocouples and heat flux density (Ahlborn sensors)	$R_{si}=\frac{\left(T_{i}-T_{si}\right)}{q}$ (Section 2.3.3)	$R=\frac{\left(T_{si}-T_{se}\right)}{q}$ (Section 2.2.2)	$R_{se}=\frac{\left(T_{se}-T_{e}\right)}{q}$ (Section 2.3.3)	$R_{tot}=R_{si}+R+R_{se}$
2b	measurement	Measurement of air temperatures by thermocouples and wall surface temperatures and heat flux density by dedicated device (Hukseflux TRSYS01)	ISO 6946, Appendix A (Section 2.3.2)	$R=\frac{R_{1}+R_{2}}{2}$ where: *R*_1_ and *R*_2_ are results of measurement by dedicated device	ISO 6946, Appendix A (Section 2.3.2)	$R_{tot}=R_{si}+R+R_{se}$
3a	measurement	Spot measurement of air and surface temperatures on warm side by IR camera, on cold side by thermocouples	ISO 6946, Appendix A (Section 2.3.2)	-	ISO 6946, Appendix A (Section 2.3.2)	$R_{tot}=\frac{R_{si}\left(T_{i}-T_{e}\right)}{\left(T_{i}-T_{si}\right)}$ (Section 2.2.1)
3b	measurement	Area measurement of air and surface temperatures on warm side by IR camera, on cold side by thermocouples	ISO 6946, Appendix A (Section 2.3.2)	-	ISO 6946, Appendix A (Section 2.3.2)	$R_{tot}=\frac{R_{si}\left(T_{i}-T_{e}\right)}{\left(T_{i}-T_{si}\right)}$ (Section 2.2.1)

**Table 3 materials-14-07438-t003:** Surface thermal resistances of tested partitions depending on method of determining them.

Method of Determining Surface Thermal Resistances	*R_si_*	*R_se_*
	m^2^K/W	m^2^K/W
Computational—ISO 6946, Table 7 (Section 2.3.1)	0.130	0.130
Computational + measurement—ISO 6946, Appendix A	0.088	0.121
Measurement—Wall A	0.101	0.099
Measurement—Wall B	0.159	0.100
Measurement—Wall C	0.169	0.136
Measurement—Wall A (insulated)	0.083	0.153
Measurement—Wall B (insulated)	0.214	0.134
Measurement—Wall C (insulated)	0.132	0.139
Measurement—mean value	0.143	0.127

**Table 4 materials-14-07438-t004:** Air and surface temperatures for tested walls by means of infrared camera measurements for all three randomly selected thermograms.

Test Object	Temperature Measurements by Infrared Camera								
	Air (SP01)			Surface (SP02, SP03)			Surface (AR01)		
	Min °C	Max °C	Mean °C	Min °C	Max °C	Mean °C	Min °C	Max °C	Mean °C
Wall A	+19.8	+20.2	+20.0	+17.9	+18.6	+18.3	+18.2	+18.2	+18.2
Wall B	+19.9	+20.3	+20.1	+15.5	+16.0	+15.7	+15.3	+15.5	+15.4
Wall C	+18.4	+20.3	+19.1	+11.0	+12.5	+11.6	+11.5	+11.6	+11.5
Wall A (insulated)	+19.6	+20.2	+20.0	+19.2	+19.7	+19.4	+19.2	+19.4	+19.3
Wall B (insulated)	+20.3	+20.5	+20.4	+19.4	+19.7	+19.5	+19.1	+19.3	+19.2
Wall C (insulated)	+19.9	+20.3	+20.2	+18.9	+19.2	+19.0	+19.1	+19.2	+19.2

**Table 5 materials-14-07438-t005:** Comparison of determined total thermal resistances.

Method of Determining Total Thermal Resistance	Total Thermal Resistance, *R_tot_* m^2^K/W					
	Wall A	Wall B	Wall C	Wall A (Insulated)	Wall B (Insulated)	Wall C (Insulated)
0a	1.403	0.585	0.510	3.908	3.090	3.015
0b	1.352	0.534	0.459	3.857	3.039	2.964
1	1.440	0.411	0.330	3.750	2.480	2.435
2a	1.372	0.464	0.467	4.056	3.763	3.162
2b	1.305	0.537	0.595	3.621	3.241	2.839
3a	1.464	0.540	0.308	3.998	2.962	2.247
3b	1.410	0.508	0.307	3.798	2.139	2.547
Average	1.392	0.511	0.425	3.855	2.959	2.744
Standard deviation	0.050	0.053	0.104	0.138	0.486	0.313
Max–Min	0.159	0.173	0.288	0.435	1.624	0.915
Relative range (from average)	−6.2% to +5.2%	−19.5% to +14.3%	−27.8% to +40.0%	−6.1% to +5.2%	−27.7% to +27.2%	−18.1% to +15.2%

**Table 6 materials-14-07438-t006:** Difference between determined thermal resistances and mean value (colors explained in the text below).

Method of Determining Thermal Resistance	Difference between Given Method and Mean Value (*R_tot,k_*—*R_tot,mean_*) m^2^K/W (+ Above Mean, − Below Mean) % (Absolute Value)						
	Wall A	Wall B	Wall C	Wall A (Insulated)	Wall B (Insulated)	Wall C (Insulated)	Average (Without One Worst Result)
0a	+0.011	+0.073	+0.085	+0.052	+0.130	+0.271	-
	0.8	14.3	19.9	1.4	4.4	9.9	6.1
0b	−0.040	+0.023	+0.034	+0.002	+0.080	+0.220	-
	2.9	4.4	8.0	0.0	2.7	8.0	3.6
1	+0.048	−0.100	−0.096	−0.106	−0.479	−0.309	-
	3.4	19.5	22.5	2.7	16.2	11.3	10.6
2a	−0.021	−0.047	+0.042	+0.201	+0.804	+0.418	-
	1.5	9.2	9.9	5.2	27.2	15.2	8.2
2b	−0.087	+0.026	+0.170	−0.234	+0.282	+0.095	-
	6.2	5.0	40.0	6.1	9.5	3.4	6.1
3a	+0.072	+0.029	−0.117	+0.142	+0.003	−0.497	-
	5.2	5.6	27.5	3.7	0.1	18.1	6.5
3b	+0.018	−0.004	−0.118	−0.057	−0.820	−0.198	-
	1.3	0.7	27.8	1.5	27.7	7.2	7.7

## Data Availability

Not applicable.

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
