# Peer review of "Non-Destructive Possibilities of Thermal Performance Evaluation of the External Walls"

_materials, 2021, doi:10.3390/ma14237438_

Round 1

Reviewer 1 Report

Manuscript ID: materials-1421735

Title: Non-destructive possibilities of thermal performance evaluation of the building partitions

The manuscript  presented a comparison of calculated total thermal resistance values and the measured ones for three kinds of masonry walls without and with thermal insulation using three methods. Overall, this manuscript is within the scope of the journal and provides fair information about experimental work. However, there are some minor issues that should be addressed.

Comments:

1- It is better to give a reason about why you selected these three wall elements in this paper.

2- Authors should use a consistent temperature unit. Now you sometimes used Celsius degree and sometimes used Kelvin degree.

3- Abstract is too qualitative. It is recommended that the authors include some key quantitative results in the abstract.

4- It is better to use the same temperature scale for a and b in Figs 6, 7, and 8.

5- Add standard deviation beside the average in Table 5.

6- The legends are not legible in Figs 11, 12 and 13.

7- In conclusion, there are lots of paragraphs that should be integrated.

Reviewer 2 Report

The research is relevant and well developed.

The methodology is clear and well structured.

The results are sufficiently well justified and clearly stated.

However, all the research is established from the postulates of the current regulations without questioning whether the thermal reality to which the buildings are exposed is comparable to the assumptions tested or raised in the regulations. 

It is not appropriate to refer to building envelopes as partitions. If the assessment of the building envelope is being analyzed, the term envelope should be used. From this point of view, the title should be modified .

Proposed title: Non-Destructive Possibilities of Thermal Performance Evaluation of the Building Envelopes

It would be desirable some kind of commentary from the authors about how climatological agents such as rain, wind or radiation affect the surface of the cases studied.

Reviewer 3 Report

1) I advise using "building constructions" instead of "building partitions". 

2) Too many keywords, you should cut some.

3) In the introduction, the first chapter should be deleted. I read this first chapter at least a thousand times in papers... no need for it here. 

4) Mass citations in the introduction chapter. The authors did mass citations (e.g. 12-21, 22-26, 34-41, etc.), which should be extended a bit broken into smaller groups. 

5) There are few key papers missing from recent years from the references and introductions dealing with in-situ measurements or measurement methods of building constructions. Among others, e.g. 

- https://doi.org/10.1016/j.enbuild.2018.10.021 - https://doi.org/10.1016/j.csite.2021.100941 - https://doi.org/10.1016/j.enbuild.2019.109417 - https://doi.org/10.4028/www.scientific.net/AMM.887.605 - https://doi.org/10.3390/su6107107 - https://doi.org/10.1556/606.2017.12.3.7 - https://doi.org/10.1016/j.jobe.2020.101637

6) I also miss introducing ISO 9869-2:2018, because it describes the infrared method for in-situ testing of building constructions!

7) Instead of "thermal capacity", I advise using "heat capacity"

8) In Materials and Methods, authors start now with methods. I suggest introducing first the materials that you investigated, then the methods that you use for the experiment. The introduction of the materials is now part of the subchapter "test setup" only. 

9) If you refer to thermal conductivity what is measured, please use effective thermal conductivity, since it is not the declared value that can be obtained from technical datasheets, but an effective value that is influenced by temperature and moisture conditions during the experiment.

10) Introduction of ISO 6946 surface resistance calculation (subchapter 2.3.1 and 2.3.2) could be deleted and only referenced as the used method for calculating the surface resistances! It is described in the standard in detail, no need to repeat it in the paper.

11) I also advise that after you introduced the materials, then you introduce the test setup with the climate chambers and sensors, and then you include the methods to calculate the thermal transmittance of building constructions according to the measurements.

12) The details of building enclosure preparation for testing could be detailed, e.g. how many days you waited after building the enclosures before testing? What conditions were in the laboratory where the walls are made? How do you make sure that construction moisture does not affect the measured values? 

13) I think the thermal conductivities in Table 2 are declared values and not design values. Sources could be added (now there is only a sentence, which does not declare what kind of material specification tables were used, they are not referenced). 

14) Table 3 is a very good summarizing table, which also summarizes the earlier included equations. Therefore, I think no need to show equations twice, you could delete them from the methodology chapters... This table shows everything that is needed (except the thermal imager method).

15) Results are sometimes too detailed, e.g. you wrote the substituted numbers into equations (e.g. e.q. 17, 18) or simply show a surface resistance as an equation (e.g. 19, 20, 21, 22), while there is a table 4 which shows the results again. No need to show the results multiple times, show them in summarizing tables, and it is also better to compare them together there.

16) To show the results from thermal imagers, I also advise using summarizing tables, instead of listings with bullet points.

17) The most important table is Table 5 showing all results. The figures then show the components separately, but I think the y-axis should be improved, also the quality of the figures need to be increased.

18) In the results chapter, there are also too many bullet points, e.g to write single measured values, you use bullet points, instead of summarizing them into tables. There are also page long bullet point lists about difference percentages between results that also can be written in simple text or summarize in a table.

19) Discussing the results, I think there are too many bullet points. I advise you to simply write about the results, bullet points are for highlighting the most important information, results or conclusions. 

20) I think the current manuscript needs a major revision, but the topic and the experiments presented in the paper are worthy to publish after the text and figures are corrected as advised!

Round 2

Reviewer 1 Report

The authors have carefully addressed the reviewer’s questions, and hence the current manuscript can be considered for publishing.

Author Response

Response to Reviewer 1

Thank you for your kind words. Having our hard work in mind, we appreciate that.

Kind regards
Authors

Reviewer 3 Report

The authors addressed most of the reviewer's comments.

A few, mostly editing issues are the following:

1) I think Table 1 is not needed in the paper, since the authors write that the values they used are from ISO 6946 and they also write down the exact values in 2.3.1. The content of the table is partly redundant and not relevant.

2) In ISO 6946, only the external heat transfer coefficient is described with e.q. 14, there is no wind speed used in the internal value, instead Table C.1. is used. In ISO 6946, v is called wind speed, and not air movement velocity. It is an application of the authors to use the same equation for the internal (warm) side similarly to the external (cold) side. Please mention this in the paper.

3) It is sometimes hard to read the equations and the descriptions of the symbols, the symbols are not always right under the equations where they belong. E.g. e.q. 13 is at line 254 while the description of the wind speed (air movement velocity) in the eq. comes after two other equations not including wind speed at line 258. I suggest putting the description of the symbols right after the equation where it belongs. I advise this to eq. 6, eq. 13 and 14.

4) In eq. 4 and 5, eq. 6 or eq. 17 and 18, v or nu is used for surface temperature. It would be better to use theta or tsi/tse, since v or nu can be easily confused with v for wind speed. Please check the symbols of ISO standard to form a consistent nomenclature. In ISO standards, T is used for thermodynamic temperatures (K), theta is used for temperatures having °C unit, while indexes are used for the location, like i for internal, e for external, s for surface, etc.

5) 20 °C is 293.15 K and -10 °C is 263.15 K. 

6) Table 5: I suggest separating min and max temperature values into separate columns, instead of writing them.

7) Subchapter title 3.2 breaks into a different page, it would be better if it is not located on the bottom of page 17, but the top of page 18. 

8) please separate numbers and units, instead of writing e.g. "10.7°C" write "10.7 °C"

Author Response

Response to Reviewer 3

We thank you again for valuable comments on our article. Our response to them can be read below. We also have made the necessary improvements to the article.

1) I think Table 1 is not needed in the paper, since the authors write that the values they used are from ISO 6946 and they also write down the exact values in 2.3.1. The content of the table is partly redundant and not relevant.

The goal in this article was to present our research in a stand-alone format. In this context the Table 1 is needed in the section 2.3.1, because it is called in the formula (1) and the section which contains is mentioned in the Table 3 -  the one which has all of most important assumptions of our research. The carried research can be repeated for other kind of partitions like roofs or floors, and now there is all of the data needed to perform that. Taking all that into account we would like to leave the Table 1 in the updated paper.

2) In ISO 6946, only the external heat transfer coefficient is described with e.q. 14, there is no wind speed used in the internal value, instead Table C.1. is used. In ISO 6946, v is called wind speed, and not air movement velocity. It is an application of the authors to use the same equation for the internal (warm) side similarly to the external (cold) side. Please mention this in the paper.

This is a valuable comment. This information is now added to the paper.

3) It is sometimes hard to read the equations and the descriptions of the symbols, the symbols are not always right under the equations where they belong. E.g. e.q. 13 is at line 254 while the description of the wind speed (air movement velocity) in the eq. comes after two other equations not including wind speed at line 258. I suggest putting the description of the symbols right after the equation where it belongs. I advise this to eq. 6, eq. 13 and 14.

This is a valuable comment. We have changed the placement of the descriptions.

4) In eq. 4 and 5, eq. 6 or eq. 17 and 18, v or nu is used for surface temperature. It would be better to use theta or tsi/tse, since v or nu can be easily confused with v for wind speed. Please check the symbols of ISO standard to form a consistent nomenclature. In ISO standards, T is used for thermodynamic temperatures (K), theta is used for temperatures having °C unit, while indexes are used for the location, like i for internal, e for external, s for surface, etc.

This is a valuable comment. We have renamed the temperatures according the ISO 7345 standard.

5) 20 °C is 293.15 K and -10 °C is 263.15 K. 

This is a valuable comment. We now have it improved in the updated paper.

6) Table 5: I suggest separating min and max temperature values into separate columns, instead of writing them.

This is a valuable comment. We have added columns the Table 5.

7) Subchapter title 3.2 breaks into a different page, it would be better if it is not located on the bottom of page 17, but the top of page 18. 

We see the subchapter title properly - on the top of page 18, so probably the paper version which is send for reviewing has a different page setup. Thank you for pointing that out, we will check page layout before publishing.

8) please separate numbers and units, instead of writing e.g. "10.7°C" write "10.7 °C"

This is a valuable comment. We now have it corrected in the paper.

Kind regards
Authors